# Characterization of Two New Shiga Toxin-Producing *Escherichia coli* O103-Infecting Phages Isolated from an Organic Farm

**DOI:** 10.3390/microorganisms9071527

**Published:** 2021-07-17

**Authors:** Yujie Zhang, Yen-Te Liao, Alexandra Salvador, Valerie M. Lavenburg, Vivian C. H. Wu

**Affiliations:** Produce Safety and Microbiology Research Unit, U.S. Department of Agriculture, Agricultural Research Service, Western Regional Research Center, Albany, CA 94710, USA; yujie.zhang@usda.gov (Y.Z.); yen-te.liao@usda.gov (Y.-T.L.); alexandra.salvador@usda.gov (A.S.); valerie.lavenburg@usda.gov (V.M.L.)

**Keywords:** STEC O103 strain, STEC-specific bacteriophage, biocontrol agents, whole-genome sequencing

## Abstract

Shiga toxin-producing *Escherichia coli* (STEC) O103 strains have been recently attributed to various foodborne outbreaks in the United States. Due to the emergence of antibiotic-resistant strains, lytic phages are considered as alternative biocontrol agents. This study was to biologically and genomically characterize two STEC O103-infecting bacteriophages, vB_EcoP-Ro103C3lw (or Ro103C3lw) and vB_EcoM-Pr103Blw (or Pr103Blw), isolated from an organic farm. Based on genomic and morphological analyses, phages Ro103C3lw and Pr103Blw belonged to *Autographiviridae* and *Myoviridae* families, respectively. Ro103C3lw contained a 39,389-bp double-stranded DNA and encoded a unique tail fiber with depolymerase activity, resulting in huge plaques. Pr103Blw had an 88,421-bp double-stranded DNA with 26 predicted tRNAs associated with the enhancement of the phage fitness. Within each phage genome, no virulence, antibiotic-resistant, and lysogenic genes were detected. Additionally, Ro103C3lw had a short latent period (2 min) and a narrow host range, infecting only STEC O103 strains. By contrast, Pr103Blw had a large burst size (152 PFU/CFU) and a broad host range against STEC O103, O26, O111, O157:H7, and *Salmonella* Javiana strains. Furthermore, both phages showed strong antimicrobial activities against STEC O103:H2 strains. The findings provide valuable insight into these two phages’ genomic features with the potential antimicrobial activities against STEC O103.

## 1. Introduction

Bacteriophages (or phages) are viruses that infect bacteria and are the most abundant biological entities in the biosphere, with estimated numbers of 10^31^ virions [1]. Phages are widely present in different environments, such as the global ocean, lake, and agricultural soil, and can infect their bacterial hosts as natural predators [2,3]. Based on different lifecycles, bacteriophages are classified into lytic phages and lysogenic phages [4]. Lytic phages lyse bacterial hosts and produce phage progenies, whereas lysogenic phages enable the phage DNA to be integrated into the bacterial chromosome without immediate bacterial lysis during the lysogenic cycle [5]. Therefore, phages play an important role in shaping the ecology and evolution of microorganism communities through phage-host interactions [6,7].

In recent years, the emergence of antibiotic-resistant pathogenic bacteria has become a crucial health problem worldwide. Lytic phages have been considered as potential antimicrobial agents to reduce antibiotic-resistant issues [8,9]. Phage-based biocontrol of bacterial pathogens has been applied in several areas, including food processing, agriculture, and aquaculture [10,11,12]. Phages should have several biological and genetic features to be considered suitable biocontrol candidates, such as a broad host range and free of virulence, antibiotic-resistant, and lysogenic genes. Therefore, a thorough characterization of the phages plays a crucial role in screening potential phages and confirming safety before phage application.

Shiga toxin-producing *Escherichia coli* is one of the major foodborne pathogens that produce Shiga toxins and can cause severe human illness, such as hemolytic-uremic syndrome (HUS) [13]. STEC strains cause approximately 1,904,891 illnesses from 2016–2018 in the United States, with 35,058 hospitalizations and 1956 deaths, according to the National STEC surveillance data from the Centers for Disease Control and Prevention (CDC) [14]. The most common STEC is O157:H7, associated with multistate outbreaks of the United States, such as the romaine lettuce outbreak in 2018 [15]. In addition, non-O157 STEC serotypes such as O26, O45, O111, O121, O145, and O103, in particular, have been recognized as a growing public health concern [14]. In 2019 and 2020, the CDC reported that *E. coli* O103 is a major serotype known to cause three outbreaks related to the contaminated ground beef and clover sprouts in the United States [16,17]. Additionally, in 2017, a foodborne outbreak in Germany was linked to the consumption of *E. coli* O103:H2-contaminated raw cow milk during a school trip to Austria, and 45 of 200 students and teachers developed gastroenteritis [18]. Several studies demonstrated that *E. coli* O103 was the most common serogroup next to *E. coli* O157, causing human illnesses [18,19]. Although several traditional interventions have been commonly used in the food industry to control foodborne pathogens, there is still a sustained global increase in foodborne outbreaks related to non-O157 STEC pathogens [20,21,22]. Thus, phage application, considered as a novel approach in the agricultural field, could offer a potential solution to control *E. coli* O103 contamination and improve food safety.

Bhages are natural bacterial antagonists, and their biocontrol potential to prevent the spread of foodborne pathogens has been revisited [23]. Several commercial phage products are available on the market to control *Listeria monocytogenes*, *Salmonella* spp., and *E*. *coli* O157:H7 through direct application on foods or in food production environments [24,25,26,27,28]. Compared to common chemical and physical treatments, lytic phages’ antimicrobial features could tackle some drawbacks of these traditional intervention technologies. For example, phages can co-evolve with target pathogens, reducing antimicrobial resistance, and the quality impact on the phage-treated food products could be minimal [29,30]. A number of phages against STEC O157:H7 were isolated and characterized in several studies [31,32]. Two research groups isolated phages phiC119 and AKFV33, both targeted specifically against STEC O157 strains, were evaluated their biocontrol potential, including host range and lysis time, and identified the presence of harmful genes [33,34]. We previously isolated and characterized a lytic bacteriophage vB_EcoS-Ro145clw with antimicrobial activity against STEC O145 and *E. coli* O145:H28 outbreak strains [35]. We also investigated genomic characteristics of the phage contributing to phage stability and antimicrobial activity for a broad phage application in various environments. Although there are many isolated phages lytic against the top 6 non-O157 STEC, information regarding the characterization of STEC O103-infecting phages is still lacking. Therefore, the objective of this study was to characterize two new STEC O103-infecting phages isolated from an organic farm, using biological and genomic approaches to explore their antimicrobial potential.

## 2. Materials and Methods

### 2.1. Bacterial Strains

A total of 17 *E. coli* strains were selected as host strains for phage isolation, and additional 17 *Salmonella* strains were used for determining the phage host ranges. The information of all strains used in this study was listed in Table 1. Briefly, a loopful of each strain was inoculated in a sterile 10 mL tryptic soy broth (TSB; Difco, Becton Dickinson, Sparks, MD, USA) and incubated overnight at 37 °C with shaking at 90 rpm prior to use.

### 2.2. Bacteriophage Isolation and Purification

Phage Ro103C3lw was isolated from non-fecal compost, and phage Pr103Blw was previously isolated from bovine feces [36]. The isolation and purification of phages were performed as previously described [37]. Briefly, the isolated phages, which were targeting STEC O103:H2 strains (RM13322 and RM10744), were subjected to phage purification process using a single-plaque purification method. Subsequently, an aliquot of 50 μL isolated phages was enriched with 100 μL of overnight STEC O103:H2 culture (RM10744) in 40 mL of TSB with 10 mM of CaCl_2_ and incubated at 37 °C overnight. The enriched phages were filtered using sterile 0.22-μm membrane filters, followed by concentration via 50 kDA cut-off Amicon Ultra-15 Centrifugal Filter Units (Merck Millipore, Billerica, MA, USA). Furthermore, cesium chloride (CsCl) gradient was used to purify the phages before conducting morphology classification under a microscope and whole-genome sequencing.

### 2.3. Biological Characteristics

#### 2.3.1. Transmission Electron Microscopy

The CsCl-concentrated phage (6 µL) was added on a copper mesh grid (Ted Pella Inc., Redding, CA, USA) and incubated at room temperature for 1 min before negative staining for 30 s via adding 0.75% uranyl acetate (Sigma-Aldrich, Darmstadt, Germany). The grids were subsequently used to observe phage morphology using a transmission electron microscope (Tecnai G2 F20 model FEI, USA).

#### 2.3.2. One-Step Growth Curves

One-step growth curve experiments were conducted as previously described with subtle modification [33]. The host strain of *E. coli* O103:H2 (RM10744) was grown in 20 mL TSB at 37 °C to reach optical density at 600 nm (OD_600_) of 0.5. Later, phage (Ro103C3lw or Pr103Blw) was added to the bacterial culture at a multiplicity of infection (MOI) of 0.01, and incubated at room temperature (5 min) for phage adsorption. After centrifuging at 10,000× *g* for 5 min at 4 °C to remove supernatant, the bacterial pellet was washed three times with 1 mL of TSB and subsequently resuspended in 20 mL TSB. The resuspended culture was further 100-fold diluted in 20 mL TSB and incubated at 37 °C with sharking at 90 rpm throughout the entire experiment. Upon culture resuspension, phage-infected bacterial cell counts were determined by mixing 10 μL of the sample (diluted culture) with 500 μL of the overnight culture of *E. coli* O103:H2 and 3 mL of molten 50% TSA agar prior to pouring into a pre-poured 18-mL TSA plate. During the incubation, 1 mL of sample was collected at a 2 min interval for phage Ro103C3lw (for a total of 20 min) and a 5 min interval for phage Pr103Blw (for a total of 55 min). The sample from each time point was flited through a sterile 0.22-μm membrane filter for double-layer plaque assay to determine phage latent periods using the similar protocol, as described previously [35]. The plates were incubated at 37 °C for 24 h. The one-step growth curve experiment of each phage was performed three times.

### 2.4. Antimicrobial Activities

#### 2.4.1. Host Range

After phage purification, phage Ro103C3lw and Pr103Blw were subjected to host range test against three generic *E. coli*, 14 STEC strains, including the serogroups of O157 and the top six non-O157, and 17 *Salmonella* strains (Table 1) using the spot test assay, as previously described [37].

#### 2.4.2. Bacterial Challenge Assay

The bacterial challenge assay was performed to measure bacterial growth treated with different MOIs of phages, as previously described, with minor changes [28]. Bacterial cultures of *E. coli* O103:H2 strains (RM13322 and RM10744) were prepared and diluted to the concentration of 1 × 10^6^ CFU/mL. Two bacterial cultures were mixed as the bacterial cocktail for the bacterial challenge assay. A measure of 200 µL of the bacterial cocktail per well was added into a 96-well plate; then, 10 µL of individual phage (Ro103C3lw or Pr103Blw) was added to each well containing the bacterial cocktail to reach MOIs of 10, 100, and 1000, accordingly. The plate was placed in a spectrophotometer (BioTek, Winooski, WT, USA) at 37 °C, and the OD_600_ reading was measured every 30 min for 18 h.

### 2.5. Genomic Characteristics

#### 2.5.1. Phage DNA Extraction and Whole-Genome Sequencing

Phage DNA was extracted using a Norgen Biotek phage DNA extraction kit (Thorold, ON, Canada) and further used to prepare phage DNA library for sequencing using a MisSq Reagent Kit v2 (500-cycle) on Illumina MiSeq sequencer (Illumina, San Diego, CA, USA) according to the manufacturer’s instruction. Approximately 6 million 2 × 250 bp pair-end sequence reads were generated for each phage. Furthermore, the genome assembly and annotation were performed using the processes as previously described [38]. Briefly, quality reads were obtained after checking raw sequence reads using FASTQC and trimming using Trimmomatic with the setting of Q30 [39,40]. The resulting quality reads were subjected to de novo assembly using Unicycler Galaxy v0.4.6.0 (SPAdes) and annotation via Prokka v1.12.0 (https://github.com/tseemann/prokka; accessed on 10 November 2020) with default parameters. Subsequently, the annotation was manually confirmed using Geneious (v11.0.3, Biomatters, New Zealand) based on the results of blastp against the National Center for Biotechnology Information (NCBI) database. The predicted tRNAs in the phage genome were confirmed using tRNAscan-SE Search Server [41]. PhageTerm was used to predict the termini and the phage DNA packaging mechanisms [42]. In addition, the screening of virulence genes and antibiotic resistance genes in the phage genome was conducted via Virulence Finder v2.0 (https://cge.cbs.dtu.dk/services/VirulenceFinder/; accessed on 10 November 2020) and ResFinder v3.0 (https://cge.cbs.dtu.dk/services/ResFinder/; accessed on 10 November 2020) web servers, respectively [43,44].

#### 2.5.2. Comparative Analysis

Two new phage sequences (Ro103C3lw and Pr103Blw) were used for blast search against the NCBI nucleotide database to obtain reference phage genomes, sharing high nucleotide similarity to each of the phages. The complete genomes of five phages sharing at least 85% identity and 85% coverage with phage Ro103C3lw were obtained from the NCBI database and further used as the reference genomes of phage Ro103C3lw. The reference phage genomes of phage Ro103C3lw included *Cronobacter* phage GW1 (GenBank accession #MH491167), *Citrobacter* phage SH4 (GenBank accession #KU687350), *Citrobacter* phage SH5 (GenBank accession #KU687351), *Escherichia* phage vB_Ecop-Ro45Lw (GenBank accession #MK301532), and *Escherichia* phage Pisces (GenBank accession #MK903277). The complete genomes of five phages sharing at least 95% identity and 95% coverage with phage Pr103Blw were obtained from the NCBI database and further used as the reference genomes of phage Pr103Blw. The reference phage genomes of phage Pr103Blw included *Escherichia coli* O157 typing phage 12 (GenBank accession #KP869110), *Escherichia coli* O157 typing phage 11 (GenBank accession #KP869109), *Enterobacteria* phage wV8 (GenBank accession #EU877232), *Escherichia* phage JN01 (GenBank accession #MN882542), and *Escherichia* phage vB_EcoM-Ro111lw (GenBank accession #MH571750). The comparison of genome maps between phage Ro103C3lw and Pr103Blw and their reference genomes was visualized using BLAST Ring Image Generator (BRIG) with the default settings visualization tool [45]. Comparative analysis of the genes related to phage infection, lysis, and packaging was performed with the MAFFT multiple sequence alignment using Geneious (v11.1.5, Biomatters, New Zealand) [46]. The phylogenetic tree was constructed with MEGA X with the maximum composite likelihood method [47].

#### 2.5.3. Nucleotide Sequence Accession Numbers

The complete genome sequences of phages vB_EcoP-Ro103C3lw and vB_EcoM-Pr103Blw have been deposited in GenBank under the accession numbers of MN067430 and MW481326, respectively.

## 3. Results

### 3.1. Biological Characterization of Phages

For morphological classification, phage Ro103C3lw belonged to the *Podoviridae* family; it had a head size with approximately 61.1 ± 0.5 nm in diameter and 58.8 ± 0.5 nm in length, and a short tail with 17.0 ± 0.5 nm in length and 23.5 ± 0.5 nm in diameter (Figure 1a). Phage Pr103Blw had the morphology belonging to the *Myoviridae* family and contained a capsid with approximately 75.0 ± 0.5 nm in diameter and 73.6 ± 0.5 nm in length; its long contractile tail was about 118.8 ± 0.5 nm in length and 21.0 ± 0.5 nm in diameter (Figure 1b).

Additionally, the one-step growth curve was conducted against *E. coli* O103:H2 strain (RM10744) for both phages to evaluate their growth factors. The results showed that a complete lytic cycle for phage Ro103C3lw was 18 min, with a latent period of 2 min, and for Pr103Blw was 45 min, with a latent period of approximately 15 min (Figure 2). Additionally, phages Ro103C3lw and Pr103Blw had a burst size of 18 and 152 PFU per infected cell, respectively (Figure 2).

### 3.2. Host Range of the Phages

The host range results revealed that phage Ro103C3lw had a narrow host range, specific to STEC O103 strains, and produced a lysis zone against the selected STEC O103:H2 strains (RM13322 and RM10744) (Table 1). By contrast, phage Pr103Blw had a broad host range, showing strong lysis against STEC O103:H2 (RM13322 and RM10744), STEC O26:H- (RM18132), and *Salmonella* Javiana strains, medium lysis against STEC O111:H- (RM11765), and weak lysis against STEC O26:H- (RM17133), STEC O157:H7 (RM18959 and ATCC 35150), and generic *E. coli* DH5α strains (Table 1).

### 3.3. Antimicrobial Activity against STEC O103 Pathogens

The antimicrobial effects of phages Ro103C3lw and Pr103Blw were determined using different MOIs against a two-strain cocktail of *E. coli* O103:H2 (RM13322 and RM10744) via a spectrophotometer (Figure 3). The results showed that the control group (bacterium without phage) started to grow after 2.5 h of incubation at 37 °C. On the contrary, no bacterial growth in all groups treated with phage Ro103C3lw, regardless of MOIs (10, 100, and 1000), was observed throughout the entire experiment period (18 h) (Figure 3a). For phage Pr103Blw, the treated bacteria with different MOIs (10, 100, and 1000) were suppressed until 8 h of incubation at 37 °C (Figure 3b). From 8 to 16 h of incubation, the groups with MOIs of 10 and 100 had better bacterial inhibition than MOI of 1000.

### 3.4. General Genomic Characterization

Phage Ro103C3lw had a double-stranded DNA with a genome size of 39,389 bp and an average GC content of 52.8% (Figure 4a). Genome analysis showed 51 coding DNA sequences (CDSs), of which 25 encoded the proteins with hypothetical functions, and 26 were annotated with the predicted functions associated with phage morphogenesis, DNA packaging, DNA regulation and replication, and host cell lysis (Appendix A). Among the 26 CDSs, a total of 10 CDSs encoded the proteins related to the phage morphogenesis, including head-to-tail joining protein, capsid proteins, internal virion proteins, and tail fiber proteins. Three CDSs encoded endolysin, holin protein, and endopeptidase Rz responsible for host cell lysis and the release of propagated phage progenies. Additionally, phage Ro103C3lw contained CDSs coding for the proteins associated with phage DNA regulation and replication, including DNA polymerase, RNA polymerase, DNA ligase, helix-destabilizing protein, helicase, and putative HNS binding protein. In the phage Ro103C3lw genome, five CDSs annotated with the predicted functions, including phage endonucleases, DNA packaging protein, and terminases, were found in charge of the phage DNA packaging. Furthermore, the PhageTerm analysis showed that the phage genome of Ro103C3lw contained two 191-bp direct terminal repeats (DTR), with the DNA packaging mechanism belonging to DTR phage. Most of all, no harmful genes (virulence genes, antibiotic-resistance genes, and lysogenic genes) were found in the phage Ro103C3lw.

Phage Pr103Blw had an 88,421-bp double-stranded DNA and an average GC content of 38.7% (Figure 4b). A total of 131 CDSs and 26 tRNAs were annotated in the genome of Pr103Blw, and 44 CDSs were predicted with the known functions (Appendix A). Specifically, a total of 19 CDSs with the predicted functions were related to phage morphogenesis, including capsid protein, membrane proteins, tail assembly proteins, tail proteins, tail fiber proteins, and tape measure chaperones. Two CDSs annotated with the functions of lysin and holin proteins, attributed to host cell lysis, were found in the phage Pr103Blw genome. There were 11 CDSs with the predicted functions regarding phage DNA regulation and replication, such as dihydrofolate reductases, DNA polymerase, and DNA helicase. The CDSs predicted with the functions of terminase and peptidase were associated with the phage DNA packaging. Furthermore, the PhageTerm results demonstrated that phage Pr103Blw contained the DNA packaging mechanism of DTR, with two 581-bp terminal repeats. No lysogenic genes, virulence genes, and antibiotic-resistance genes were identified in the phage genome.

### 3.5. Comparative Analysis of Phage Ro103C3lw

The blastn result showed that Ro103C3lw had the highest nucleotide sequence similarity (87.78% identity and 86% coverage) to *Cronobacter* phage GW1. In addition, the Ro103C3lw genome also shared a minimum of both 85% nucleotide identity and coverage with the genomes of *Citrobacter* phage SH4, *Citrobacter* phage SH5, *Escherichia* phage Pisces, and *Escherichia* phage vB_EcoP-Ro45lw. The Bacterial and Archaeal Viruses Subcommittee (BAVS) of the International Committee on Taxonomy of Viruses (ICTV) indicates that the threshold for species-level classification is at least 95% nucleotide sequence identity based on a blastn search [48]. Thus, phage Ro103C3lw was taxonomically identified as a novel species belonging to the *Kayfunavirus* genus of *Autographiviridae* family (which is derived from *Podoviridae* family) under the order of *Caudovirales*. Comparative analysis showed that phage Ro103C3lw and its five closely related reference phages, as stated above, shared most of the CDSs with known function; however, two CDSs with the respective functions of host RNA polymerase inhibitor and putative tail fiber in the Ro103C3lw genome were absent in the genomes of all reference phages (Figure 5a). Furthermore, phylogenetic analyses showed that phage Ro103C3lw contained some CDSs with the predicted functions related to phage infection (tail fibers) and host cell lysis (lysin, holin, and spanin) that were genetically different from the counterparts in these five reference phages (Figure 5b,c). On the contrary, Ro103C3lw had two predicted CDSs coding for terminases, associated with phage DNA packaging, showing a close evolutionary relationship with that of *Escherichia* phage Pisces (Figure 5d).

### 3.6. Comparative Analysis of Phage Pr103Blw

The blastn results showed that the taxonomy of myophage Pr103Blw belonged to the *Felixounavirus* genus under the order *Caudovirales*. The complete genome of phage Pr103Blw shared 97.28% nucleotide sequence identity over 95% coverage with the genome of phage *Escherichia coli* O157 typing phage 12. Comparative analysis showed that phage Pr103Blw contained most CDSs with the predicted functions similar to the counterparts in the five closely related reference phages (*Escherichia coli* O157 typing phage 12, *Escherichia* phage JN01, *Escherichia* phage vB EcoM-Ro111lw, *Escherichia* phage wV8, and *Escherichia coli* O157 typing phage 11) obtained from the NCBI database; however, one CDS associated with tail fiber was different from that of these five reference phages (Figure 6a). Phylogenetic analysis showed that phage Pr103Blw was closely related to the *Escherichia* phage JN01 with regard to the CDSs with the predicted function of tail fibers, which were responsible for the phage infection (Figure 6b). The CDSs of lysin and holin, both associated with hosts lysis, in the Pr103Blw genome showed a high nucleotide similarity to the counterparts in *Escherichia* phage JN01 and *Escherichia* phage vB EcoM-Ro111lw (Figure 6c). Furthermore, phage Pr103Blw contained the CDS of terminase, sharing high nucleotide sequence similarity with *Escherichia coli* O157 typing phage 12, *Escherichia* phage wV8, and *Escherichia coli* O157 typing phage 11 (Figure 6d).

## 4. Discussion

In this work, two newly isolated phages—Ro103C3lw and Pr103Blw—were characterized to evaluate the biologic and genomic features associated with antimicrobial activities, indicating their biocontrol potential against STEC O103 strains.

Short replication time is one of the primary strategies for phage to respond to external pressures, such as high bacterial densities (10^8^ CFU/mL) [49,50,51]. In the present study, the T7-like phage Ro103C3lw has a very short latent period of 2 min at a low host density of 10^5^ CFU/mL. Thus, it is necessary to investigate the specific genomic features contributing to the rapid latent period of phage Ro103C3lw. For phage progenies to be released from Gram-negative bacteria during the lytic cycle, there are three steps associated with cell lysis, from inside out, for each layer of the bacterial membrane—inner membrane, peptidoglycan, and outer membrane—regulated by the genes encoding holin, endolysin, and spanin [52]. Generally, holin protein begins to create pores on the cytoplasmic membrane and facilitate the endolysin protein to reach the peptidoglycan layer of bacterial cell walls for degradation; subsequently, spanin complex, including o-spanins and i-spanins, is responsible for disrupting the outer membrane, causing the release of progeny virions [52,53,54]. The blastn results revealed that these three genes within the phage Ro103C3lw genome shared the highest nucleotide similarity with the counterpart genes in *Cronobacter* phage Dev2 (data not shown). However, the time required for phage Ro103C3lw to complete a lytic cycle is shorter than phage Dev2 (35 min for a lytic cycle, with a latent period of 15 min) [55]. The differences of latent periods between phage Dev2 and Ro103C3lw indicate that there may be other factors regulating the short lytic cycle of Ro103C3lw. A previous study demonstrated that bacteriophage T7-coding proteins had an internal regulatory network with each other via protein—protein interactions that could affect protein function and phage features [56]. Nguyen et al. further confirmed that after phage infection, the lysis time of T7 phage was associated with the expression of T7 transcription terminator Tφ downstream gene products—gene 0.5, gene 0.6, and gene 0.7—encoding for the predicted hypothetical proteins [57]. However, the nucleotide sequences of Tφ and the related regulatory genes in bacteriophage T7 were not identified in the Ro103C3lw genome (data not shown). Therefore, other regulatory factors attributed to the rapid infection of phage Ro103C3lw should be further investigated.

A huge halo surrounding each lysis zone caused by phage Ro103C3lw was observed in this study. The size of the halo increased along with the prolonged incubation time (Appendix A). Several studies demonstrated that the halo was caused by depolymerase enzymes, which were encoded in most short-tail phages and could degrade bacterial polysaccharides, including capsular polysaccharides (CPS), exopolysaccharides (EPS), or lipopolysaccharide (LPS) [58,59,60,61,62]. These phage depolymerases were identified either as structural components of phage particles or soluble proteins generated during host cell lysis [58]. According to Pires et al., most phage depolymerases were highly associated with phage structural proteins, such as tail fibers and baseplates [59]. Phage tail fiber proteins contain two domains: the N-terminal domain is related to the attachment of phage tails, while the C-terminal domain is in charge of recognition and adsorption of LPS on the host membranes and is subsequently responsible for both phage host range and the depolymerases activity [63]. Previous studies reported that the N-terminal domain of T7 tail fiber (also called p17_T7_) was highly conserved among different T7-like phages. However, the C-terminal domain of T7 tail fiber had the enzymatical activity to degrade bacterial polysaccharides and, thus, exhibited high sequence diversity due to their polysaccharide substrate specificity [53,62]. T7-like phages K1F and L1 contained tail fiber proteins with the depolymerases activity to degrade K1 capsules of *E. coli* strains and EPS of *E. amylovora* strains, respectively [64,65]. Additionally, the tail fibers of both phages K1F and L1 harbored a conservative gp17_T7_-like N-terminal domain but a diverse C-terminal domain that only interacted with the CPS or EPS of their specific hosts. A similar result observed in the present study showed that the gene coding for tail fiber protein of phage Ro103C3lw, with depolymerase potential, shared high genetic similarity with the N-terminal of tail fibers of phage K1F, T7, and L1. However, the C-terminus of Ro103C3lw tail fiber had heterogeneous nucleotide sequences with those of phage tail fiber with depolymerases activity, likely due to their specific host range (data not shown). Moreover, several studies indicated the in vitro antimicrobial activity and anti-biofilm potential of phage depolymerases to degrade capsular polysaccharides on bacteria [66,67]. These findings reveal the diversity of phage-derived depolymerases and their enzymatic activity as a promising antimicrobial agent for the development of phage-based interventions.

Host range is one of the critical factors related to phage antimicrobial activity and is likely to be affected by the phage tail morphology. The present result showed that T7-like phage Ro103C3lw was specific to STEC O103 strains, whereas myophage Pr103Blw demonstrated a broad lytic spectrum against diverse STEC serotypes as well as *Salmonella* Javiana strains. Similarly, Korf et al. isolated the phages infecting *Escherichia coli* with different morphologies from various sources and found that the short-tail phages had a narrow host range, but those phages with a long tail had a broader host range [68]. Another study also showed that four myophages (SA35RD, SA79RD, SA20RB, and SA21RB) isolated from cattle feces displayed antimicrobial activities against more than 10 serotypes of non-O157 *E. coli* strains [69]. Phage tail structure consists of several parts, such as tail fibers, tail tube, tail spikes, and baseplate, and it is different between phage morphologies. Myophages have a long and contractile tail, while T7-like phages have a short and non-contractile tail [70]. A review conducted by Nobrega et al. summarized different mechanisms regarding the interaction of tailed phages with the phage receptors on the surface of bacterial hosts through specific tail structures [71]. Briefly, myophages could move around the bacterial cell surface through reversible binding of the specific host receptors with six extended long tail fibers until phages found an optimal site for irreversible absorption and subsequent genome ejection. However, the infection of T7-like phages involves a conformational change in the tail fibers to become perpendicular to the cell surface and then to trigger an opening of the internal tail channel prior to DNA ejection. These structural determinants between different phage types affect the phage-host interactions and subsequently result in a distinct host range. In our study, a high nucleotide sequence identity (96.4%) of the gene, long-tail fiber gp37, was observed between Pr103Blw and the reference phages lytic against *E. coli* O111 and O157 strains (Figure 5). The genomic result was aligned with the host range of phage Pr103Blw, even though the lysis against *E. coli* O111 and O157:H7 strains was not as strong as *E. coli* O103. This genomic evidence reveals that phage tail structures, particularly tail fibers, are closely associated with the phage host range and the potential phage-bacterial interactions during infection.

The multiplicity of infection (MOI) is one of the critical factors that impact phage antimicrobial efficacy [72]. In the present study, the regrowth of Pr103Blw-treated bacteria with a high MOI of 1000 was faster than that of a low MOI of 10 or 100. This similar phenomenon was also observed in a study conducted by Chen et al.; where, the phage As-gz was used to target host bacteria *Aeromonas salmonicida* (MF663675) with a series of MOIs (0.01, 0.1, 1, 10), and the bacteria treated with the phage at a MOI of 10 recovered faster than the other treatment groups with lower MOIs [73]. Additionally, the findings from other studies suggested that other factors, including phage adsorption rate, bacterial densities, and bacterial defense system, in particular, could also affect phage antimicrobial activity [72,74]. Notably, several studies suggested that a high dose of lytic phages was considered as a selection pressure and could likely accelerate the response of the bacteria defense system to cause bacterial resistance to the phage infection [75,76]. Christiansen et al. investigated the emergence of bacteriophage insensitive mutants (BIMs) with different MOIs and showed that phage-sensitive strains dominated the regrowth of bacterial population (>99.8%) at low MOIs; however, phage-resistant strains (>87.8%) were dominant at high MOIs [76]. Additionally, Middelboe et al. reported that the lysis rate of sensitive bacterial cells by lytic phage is positively correlated with the development of resistant bacterial cells [77]. These findings indicate that a high MOI is not always adequate to elicit strong antimicrobial efficacy; hence, it is likely to cause the emergence of BIMs instead. Thus, selecting an optimal MOI should be taken into consideration to maximize the antimicrobial efficacy of phage application.

The current results showed that phage Pr103Blw contained 26 tRNAs, whereas no tRNA was identified in Ro103C3lw. Morgado et al. found that the presence of tRNAs might be associated with phage morphologies, with an average of 72%, 31%, and 10% among the phage genomes belonging to the *Myoviridae*, *Siphoviridae*, and *Podoviridae* families, respectively [78]. Although the role of tRNA associated with the phage’s biological functions has not been fully explored yet, most studies found the association between tRNAs and potential enhancement of the phage fitness. Bailly-Bechet et al. indicated that tRNAs in lytic phages were highly used for the translation of phages’ genes and showed higher codon usage biases in the phage genomes than in the bacterial host [79]. For example, a study conducted by Asif et al. found that bacteriophage TAC1 genome encoded 13 putative tRNAs, and the codons corresponding to those predicted tRNAs were present at a higher frequency in the phage TAC1 genome than its host *Acinetobacter baumannii* ATCC 17,978 genome [80]. The role of tRNAs with codon usage biases was further explored by Whichard et al. They reported that, among 22 tRNAs detected in the bacteriophage Felix O1 genome, 7 tRNAs contributed to the translation of phage mRNAs in the infected cells [81]. Therefore, except for the use of the host’s translation machinery, tRNAs can facilitate phage protein translation during the infection, such as the increase in protein synthesis, to attain high fitness under different environmental stresses [78,79]. These findings suggest that tRNAs play a key role in enhancing phage fitness and contributing to phage survival via the high codon usage biases of the phage genome. The underlying mechanisms regarding tRNA associated with the host-phage interaction require future investigation.

In conclusion, two new phages—Ro103C3lw and Pr103Blw—lytic against STEC O103 strains with different morphologies were characterized via biological and genomic approaches in this study. Phage Ro103C3lw, belonging to a T7-like phage, contained a short non-contractile tail and had a narrow host range, with strong antimicrobial activity infecting STEC O103. Additionally, the tail fiber of phage Ro103C3lw exhibited a unique depolymerase enzyme activity to degrade bacterial polysaccharides. The protein could be further used as an alternative antimicrobial agent alone or in combination with phage to enhance the effectiveness. Phage Pr103Blw displayed the morphology belonging to the *Myoviridae* family and contained a long and contractile tail. Phage Pr103Blw had a tail fiber with a high nucleotide sequence similarity to the counterparts in phages infecting *E. coli* O111 and O157 strains. It displayed a broad host range against diverse STEC serotypes (O26, O103, O111, and O157) and *Salmonella* Javiana. Furthermore, there were no lysogenic potential and harmful genes identified in either phage genome. These antimicrobial features of phages Ro103C3lw and Pr103Blw demonstrated their potential as alternative antimicrobial agents against pathogenic *E. coli* O103 strains. Future studies are needed to examine the biocontrol application and the optimal conditions under which these phages will be applied.

## Figures and Tables

**Figure 1 microorganisms-09-01527-f001:**
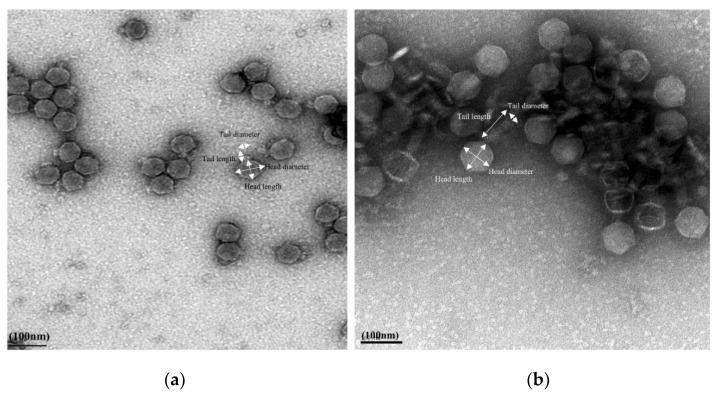
Transmission electron microscopy image of phage morphology (**a**) Ro103C3lw with a capsid (61.1 ± 0.5 nm in diameter and 58.8 ± 0.5 nm in length) and a short tail (17.0 ± 0.5 nm in length and 23.5 ± 0.5 nm in diameter), showing *Podoviridae* morphology. (**b**) Pr103Blw with a capsid (75.0 ± 0.5 nm in diameter and 73.6 ± 0.5 nm in length) and a long contractile tail (118.8 ± 0.5 nm in length and 21.0 ± 0.5 nm in diameter), showing *Myoviridae* morphology.

**Figure 2 microorganisms-09-01527-f002:**
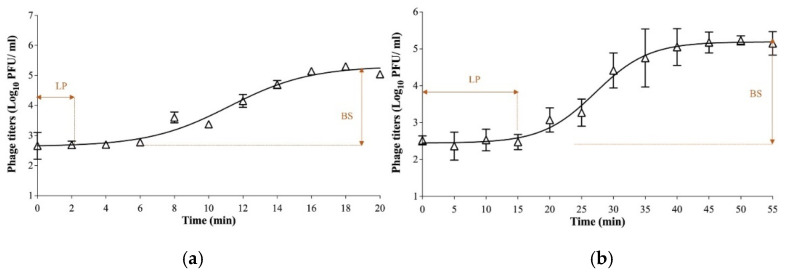
One-step growth curve of the phage Ro103C3lw (**a**) and Pr103Blw (**b**) using *E. coli* O103:H2 strain (RM10744). The growth parameters of the phage indicate a latent period (LP) and an average burst size (BS) of each phage. The error bars present the standard error of the mean for each time point of the one-step growth curve.

**Figure 3 microorganisms-09-01527-f003:**
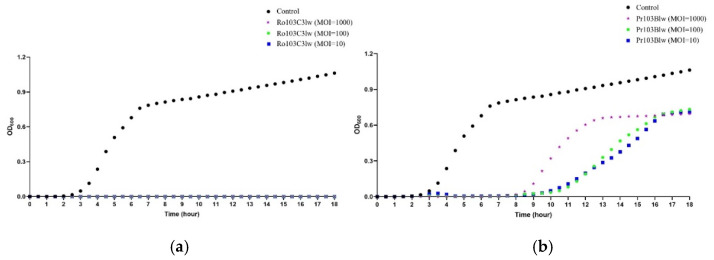
Bacterial challenge assay of the phage Ro103C3lw (**a**) and Pr103Blw (**b**) against *E. coli* O103:H2 cocktail (RM13322 and RM10744) conducted in a 96-well plate. The OD_600_ was measured by spectrophotometer every 0.5 h for the period of 18 h incubation at 37 °C. The control (black circles) contains only bacterial culture. The multiplicity of infection (MOI) of 10 (blue squares) contains bacterial culture treated with 10-fold more concentration of the phage Ro103C3lw or Pr103Blw; MOI of 100 (green circles) contains bacterial culture treated with 100-fold more concentration of the phages; MOI of 1000 (purple stars) contains bacterial culture treated with 1000-fold more concentration of the phages.

**Figure 4 microorganisms-09-01527-f004:**
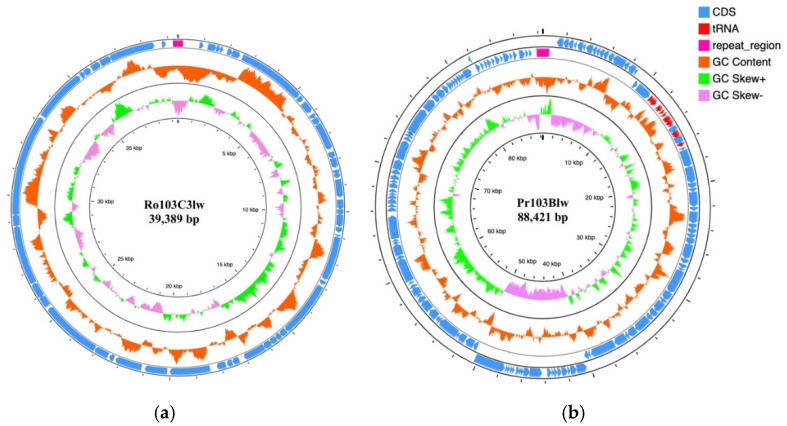
Circular genome maps of phage Ro103C3lw (**a**) and Pr103Blw (**b**) using CGview server^BETA^. The rings from inside out represent GC skew (green and light purple), GC content (orange), CDSs (blue), and repeat region (velvet). tRNAs (red) are only detected in phage Ro103Blw (**b**). The predicted functions of the annotated CDSs are listed in Appendix A.

**Figure 5 microorganisms-09-01527-f005:**
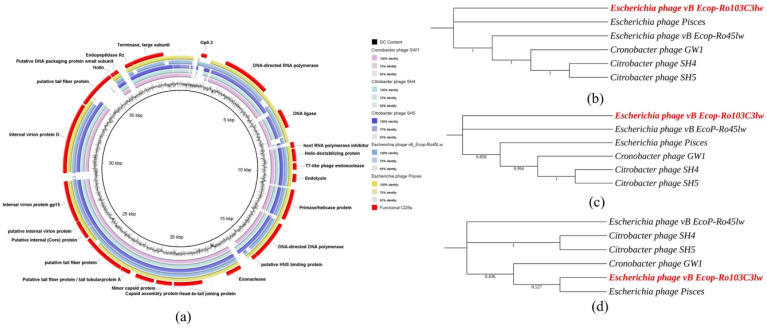
Genomic relationships between Ro103C3lw and its five closely related reference phages. (**a**) Whole-genome comparison of Ro103C3lw and five reference phages using blastn and visualization with BLAST Ring Image Generator (BRIG). The color key coding for five circular maps of reference phage genomes with high sequence identity to Ro103C3lw genome is illustrated in the legend, and the gradients of each color indicate the sequence identity, ranging from 50–100%. The CDSs with the predicted functions from the phage Ro103C3lw are displayed in the outermost ring with red color. (**b**–**d**) Maximum likelihood phylogenetic analysis of Ro103C3lw (highlighted in red) and the five closely-related reference phages based on the MAFFT alignment of the CDSs associated with (**b**) infection (tail fibers), (**c**) host cell lysis (lysin, holin, and spanin), and (**d**) DNA packaging (terminases). Numbers next to the branches are bootstrap values (500 replicates).

**Figure 6 microorganisms-09-01527-f006:**
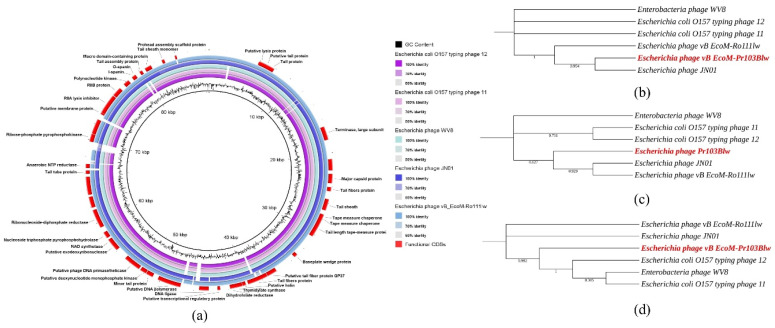
Genomic relationships between Pr103Blw and its five closely related reference phages. (**a**) Whole-genome comparison of Pr103Blw and five reference phages using blastn and visualization with BLAST Ring Image Generator (BRIG). The color key coding for five circular maps of reference phage genomes with high sequence identity to Pr103Blw genome is illustrated in the legend, and the gradients of each color indicate the sequence identity, ranging from 50–100%. The CDSs with the predicted functions from the phage Pr103Blw are displayed in the outermost ring with red color. (**b**–**d**) Maximum likelihood phylogenetic analysis of Pr103Blw (highlighted in red) and the five closely-related reference phages based on the MAFFT alignment of the CDSs associated with (**b**) infection (tail fibers), (**c**) host cell lysis (lysin and holin), and (**d**) DNA packaging (terminase). Numbers next to the branches are bootstrap values (500 replicates).

**Table 1 microorganisms-09-01527-t001:** Host range of phages Ro103C3lw and Pr103Blw against different serotypes of Shiga toxin-producing *Escherichia coli* (STEC) and *Salmonella* strains.

Host Panel	Serogroups	Bacterial Isolates	Phage		Bacterial Sources ^#^
			Ro103C3lw	Pr103Blw	
Generic *E. coli*	O157	ATCC 43888	− *	−	human feces
		ATCC 13706	−	−	n/a
		DH5α	++	+	n/a
STEC	O26	STEC O26:H- (RM18132)	−	+++	water
	O26	STEC O26:H- (RM17133)	−	+	water
	O45	STEC O45:H- (RM10729)	−	−	cattle
	O45	STEC O45:H16 (RM13752)	−	−	cattle
	O103	STEC O103:H2 (RM13322)	+++	+++	cattle feces
	O103	STEC O103:H2 (RM10744)	+++	+++	cattle feces
	O111	STEC O111:H- (RM11765)	−	++	water
	O111	STEC O111:H- (RM14488)	−	−	water
	O121	STEC O121:H19 (96-1585)	−	−	human feces ^ø^
	O121	STEC O121:H- (RM8082)	−	−	cattle feces
	O145	STEC O145:H- (RM10808)	−	−	cattle feces
	O145	STEC O145:H+ (RM9872)	−	−	cattle feces
	O157	STEC O157:H7 (RM18959)	−	+	water
	O157	STEC O157:H7 (ATCC 35150)	−	+	human feces ^ø^
*Salmonella*		*Salmonella* Agona	−	−	environment
		*Salmonella* Anatum	−	−	environment
		*Salmonella* Berta	−	−	environment
		*Salmonella* Gallinarum	−	−	environment
		*Salmonella* Infantis	−	−	environment
		*Salmonella* Javiana	−	+++	environment
		*Salmonella* Mbandaka	−	−	environment
		*Salmonella* Oranienburg	−	−	environment
		*Salmonella* Derby 45340	−	−	environment
		*Salmonella* Dublin 15480	−	−	n/a
		*Salmonella* Montevideo 51	−	−	environment
		*Salmonella* Muenster	−	−	environment
		*Salmonella* Newport	−	−	environment
		*Salmonella* Saintpaul	−	−	environment
		*Salmonella* Thompson	−	−	environment
		*Salmonella* Typhimurium ATCC 14028	−	−	chicken
		*Salmonella* Typhimurium ATCC 6962	−	−	human feces ^ø^

* indicates the degree of lysis using a cross mark, “+” indicates weak lysis, “++” indicates medium lysis, “+++” indicates complete lysis, and “–” indicates no lysis. ^#^ The bacterial sources are the type of environmental samples where the strain was originally isolated from; n/a means the information of the bacterial isolation source is not available; environment means bacteria was isolated from environmental samples, but the information about the sample type is unavailable. ^ø^ The strain was associated with foodborne outbreaks.

## Data Availability

Phages genomes reported in this study are available in GenBank under accession numbers: MN067430 (Ro103C3lw) and MW481326 (Pr103Blw).

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
