# Peer review of "Characterization of Two New Shiga Toxin-Producing Escherichia coli O103-Infecting Phages Isolated from an Organic Farm"

_microorganisms, 2021, doi:10.3390/microorganisms9071527_

Round 1
Reviewer 1 Report
The article presents the characteristics of two Shiga toxin-producing Escherichia coli O103-infecting phages isolated from an organic farm. The data are well presented and there are a few points that need attention and review. In different parts of the discussion, the authors provide a detailed discerption of previous works. I suggest the authors focus on those that are most related to the topic under discussion and describe it briefly. In addition, most of the paragraphs are too long, which may not be attractive to readers. Thus, I am hopeful that the authors amend it in the revised submission.
Specific comments
Line 48-52: The information presented in this section is based on data collected before 2011. Therefore, it would be better if the authors provided a recent estimate.
Line 218-220: As explained, the two phages were identified by authors and reported in their previous study. (29). I suggest deleting this part and directly proceed to the characterization of phages.
Line 221-224: The authors presented a brief morphological description of the phage virions based on the results of TEM (Figure 1). Figure 1b displayed a single phage virion. Assuming equal volume (6µL) aliquots of samples were used for TEM analysis, it would have been better if the authors provide TEM figure (1b) of Pr103Blw with more than one virion, as in figure 1A. In addition, it would have been better if they include information related to head length and tail diameter. Authors may refer to the following manuscript (Jurczak-kurek et al., 2016; Scientific report6: 34338).
Line 373-382: Suggest moving this section to the introduction
Line 415-449. I suggest shortening this paragraph. Authors may consider revising the outputs of previous studies explained in this section
Line 450-4: This section describes the host range and antimicrobial activity of the isolated phages in relation to morphology (tail). Therefore, the lead sentence (line 450) in this paragraph could emphasize morphology vs antimicrobial activity and/or host range.
Lines 459-461: Based on your results, both of your phages have tails (regardless of size). Therefore, it would have been better to discuss the impacts of tail size on the host range and antimicrobial activity than to explain those with and without a tail.
Line 473: "Phage tail structures, particularly tail fibers." What phage tail structures are the authors referring to? The result indicated only tail size difference between the Ro103C3lw and 519
Pr103Blw.
Line 476. A citation may be needed and suggest deleting "In addition to the host range…"
Line 519. The conclusion seems abstract and I suggest to revise.
Author Response
Manuscript ID: microorganisms-1280391
Manuscript Title: Characterization of two new Shiga Toxin-Producing Escherichia coli O103-infecting phages isolated from an organic farm
Author(s): Yujie Zhang, Yen-te Liao, Alexandra Salvador, Valerie M. Lavenburg, and Vivian C. H. Wu
Reviewer 1
The article presents the characteristics of two Shiga toxin-producing Escherichia coli O103-infecting phages isolated from an organic farm. The data are well presented, and there are a few points that need attention and review. In different parts of the discussion, the authors provide a detailed discerption of previous works. I suggest the authors focus on those that are most related to the topic under discussion and describe it briefly. In addition, most of the paragraphs are too long, which may not be attractive to readers. Thus, I am hopeful that the authors amend it in the revised submission.
Response: We thank the reviewer’s comments. We have further improved the manuscript.
Line 48-52: The information presented in this section is based on data collected before 2011. Therefore, it would be better if the authors provided a recent estimate.
Response: We thank the reviewer’s comments. We have updated the current data in the revised manuscript (lines 50-53).
Line 218-220: As explained, the two phages were identified by authors and reported in their previous study. (29). I suggest deleting this part and directly proceed to the characterization of phages.
Response: We thank the reviewer’s comments. We have updated the information based on the reviewer’s comment in the revised manuscript (lines 196-197).
Line 221-224: The authors presented a brief morphological description of the phage virions based on the results of TEM (Figure 1). Figure 1b displayed a single phage virion. Assuming equal volume (6μL) aliquots of samples were used for TEM analysis, it would have been better if the authors provide TEM figure (1b) of Pr103Blw with more than one virion, as in figure 1A.In addition, it would have been better if they include information related to head length and tail diameter. Authors may refer to the following manuscript (Jurczak-kurek et al., 2016; Scientific report6:34338).
Response: We thank the reviewer’s comments. We have updated Figure 1b in the revised manuscript. The dimensional information is also included in the updated Figure 1 and the result section.
Line 373-382: Suggest moving this section to the introduction
Response: We thank the reviewer’s comments. We have moved the information with subtle modification to the introduction in the revised manuscript (lines 68-76; 348-350).
Line 415-449. I suggest shortening this paragraph. Authors may consider revising the outputs of previous studies explained in this section
Response: We have updated the information in the revised manuscript (lines 395-409).
Line 450-4: This section describes the host range and antimicrobial activity of the isolated phages in relation to morphology (tail). Therefore, the lead sentence (line 450) in this paragraph could emphasize morphology vs antimicrobial activity and/or host range.
Response: We have updated the information in the revised manuscript (lines 410-411).
Lines 459-461: Based on your results, both of your phages have tails (regardless of size). Therefore, it would have been better to discuss the impacts on the host range and antimicrobial activity than to explain those with and without a tail.
Response: Thanks for the reviewer’s comment. In our study, phages Pr103Blw and Ro103Blw belong to entirely different families with different phage morphologies. Phages Pr103Blw belongs to Myoviridae () and phage Ro103Blw belongs to Autographiviridae ().
Based on the, the authors indicated that the tail fiber structure, varied between different phage morphologies, is a critical factor affecting the host range of a phage. Therefore, we think “tail structure from different phage morphologies, such as long and contractile-tailed phages and short and non-contractile-tailed phages” is more appropriate than “tail size.”
References:
- Nobrega, F. L., et al. (2018). "Targeting mechanisms of tailed bacteriophages." Nature Reviews Microbiology 16(12): 760-773.
- Hu, B., et al. (2015). "Structural remodeling of bacteriophage T4 and host membranes during infection initiation. Proc". Natl Acad. Sci. USA 112, E4919–E4928.
- González-García, V. A. et al. (2015). "Conformational changes leading to T7 DNA delivery upon interaction with the bacterial receptor. " J. Biol. Chem. 290, 10038–10044.
Line 473: "Phage tail structures, particularly tail fibers." What phage tail structures are the authors referring to? The result indicated only tail size difference between the Ro103C3lw and Pr103Blw.
Response: Based on the following references, ; therefore, tail structure from different phage morphologies, such as long and contractile tail, and short and non-contractile tail, is more appropriate than the term “tail size” to describe here. Additionally, the current results indicated that tail fiber was the primary part associated with the host range.
References:
- Nobrega, F. L., et al. (2018). "Targeting mechanisms of tailed bacteriophages." Nature Reviews Microbiology 16(12): 760-773.
- Hu, B., et al. (2015). "Structural remodeling of bacteriophage T4 and host membranes during infection initiation. Proc". Natl Acad. Sci. USA 112, E4919–E4928.
- González-García, V. A. et al. (2015). "Conformational changes leading to T7 DNA delivery upon interaction with the bacterial receptor. " J. Biol. Chem. 290, 10038–10044
Line 476. A citation may be needed and suggest deleting "In addition to the host range…"
Response: We thank the reviewer’s comments. We have updated the information in the revised manuscript (lines 437-438).
Line 519. The conclusion seems abstract and I suggest to revise
Response: We thank the reviewer’s comments. We have updated the information in the revised manuscript (lines 480-487).
Reviewer 2 Report
Very good research..
Author Response
We thank the reviewer's positive comments.
Reviewer 3 Report
Dear Authors
Thank you very much for your manuscript submission. Your work is well-designed and well-represented. This study is an effective reference for further investigations in future. Hence, my decision regarding this work is "Accept in present form".
Author Response

(The authors gave the same response as above.)
